# Chorioamnionitis: An Update on Diagnostic Evaluation

**DOI:** 10.3390/biomedicines11112922

**Published:** 2023-10-28

**Authors:** Sean W. D. Carter, Samantha Neubronner, Lin Lin Su, Pradip Dashraath, Citra Mattar, Sebastián E. Illanes, Mahesh A. Choolani, Matthew W. Kemp

**Affiliations:** 1Department of Obstetrics and Gynaecology, Yong Loo Lin School of Medicine, National University of Singapore, Singapore 117597, Singapore; 2Department of Obstetrics and Gynaecology, National University Hospital, Singapore 117597, Singapore; 3Center for Biomedical Research and Innovation, Reproductive Biology Program, Universidad de los Andes, Santiago 111711, Chile; 4IMPACT, Center of Interventional Medicine for Precision and Advanced Cellular Therapy, Santiago 8331150, Chile; 5Women and Infants Research Foundation, King Edward Memorial Hospital, Perth, WA 6008, Australia; 6Center for Perinatal and Neonatal Medicine, Tohoku University Hospital, Sendai 980-8574, Japan

**Keywords:** chorioamnionitis, diagnosis, sterile inflammation, biomarkers, obstetric medicine

## Abstract

Chorioamnionitis remains a major cause of preterm birth and maternal and neonatal morbidity. We reviewed the current evidence for the diagnostic tests of chorioamnionitis and how this relates to clinical practice today. A comprehensive literature search and review was conducted on chorioamnionitis and intra-uterine inflammation. Data from randomized control trials and systematic reviews were prioritized. This review highlights that sterile inflammation plays an important role in chorioamnionitis and that the current tests for chorioamnionitis including clinical criteria, maternal plasma and vaginal biomarkers lack diagnostic accuracy. Concerningly, these tests often rely on detecting an inflammatory response after damage has occurred to the fetus. Care should be taken when interpreting current investigations for the diagnosis of chorioamnionitis and how they guide obstetric/neonatal management. There is an urgent need for further validation of current diagnostic tests and the development of novel, accurate, minimally invasive tests that detect subclinical intra-uterine inflammation.

## 1. Introduction

Chorioamnionitis or intrauterine inflammation (IUI) describes any condition resulting in inflammation of either the chorion, amnion, or both. Chorioamnionitis is a major causative factor for preterm birth (birth before 37 weeks’ gestation), which is estimated to impact around 15 million pregnancies each year, and result in over 1 million deaths [1]. Chorioamnionitis is implicated in between 40 to 70% of all premature births worldwide [2,3]. Chorioamnionitis-related preterm birth, disproportionally affects the most vulnerable preterm babies, with the highest rates found in those born before 30 completed weeks of gestation [4,5]. In addition to being a risk factor for preterm birth, chorioamnionitis is independently implicated in fetal injury driven by pro-inflammatory mediators. Thus, chorioamnionitis significantly contributes to the global burden of preterm neonatal morbidity and mortality.

Traditionally, the diagnosis of clinical chorioamnionitis has been based on the signs and symptoms of patients presenting with microbial associated intrauterine inflammation and bacterial infection, first described by Gibbs et al. [6]. These criteria include maternal fever (>37.8 °C) with any two other criteria, including: maternal tachycardia (heart rate > 100 beats/min), uterine tenderness, foul-smelling amniotic fluid/vaginal discharge, fetal tachycardia (heart rate > 160 beats/min) or maternal leukocytosis (leukocyte count > 15,000 cells/mm^3^) [6,7,8,9,10]. However, new insights into the pathophysiology of chorioamnionitis have revealed a more complex picture of the relationship between clinical signs and identifiable infection. It is also reasonable to conclude that a significant degree of inflammation-associated fetal injury likely occurs in many patients prior to diagnosis with the above criteria. 

In patients with suspected chorioamnionitis, based on the clinical criteria above, intra-amniotic bacteria are detectable only 61% of the time [9], with active intra-ammonitic infection in 54% of patients and proven histopathological chorioamnionitis in only 62% of patients [11]. These data, highlight the putative role of sterile inflammation, or intrauterine inflammation without detectable bacterial invasion. Thus, illustrating that chorioamnionitis represents a large and heterogenous array of disorders marked by intrauterine inflammation that may not necessarily include the presence of bacteria within the amniotic cavity. These findings may account for why current diagnostic tests, that primarily relate to bacterial infection, have variable efficacy in accurate diagnosis of chorioamnionitis. 

This review aims to examine the current diagnostic practices for chorioamnionitis to inform clinical practice. 

### 1.1. Definitions

Due to the heterogenous nature of conditions found under the definition chorioamnionitis, different terms have been adopted in the literature to more accurately reflect the specific conditions they describe. These definitions include:

#### 1.1.1. Microbial Invasion of Amniotic Cavity (MIAC)

Positive microbial culture or PCR from amniotic fluid [7]

#### 1.1.2. Intra-Amniotic Infection (IAI)

Presence of microorganisms and inflammation within the amniotic cavity based on presence of intrauterine microbials on amniotic culture or PCR and raised amniotic fluid IL6 > 2.6 ng/mL [7].

#### 1.1.3. “Triple I” (Intrauterine Inflammation, Infection or Both)

Proposed by an expert panel on chorioamnionitis at the National Institute of Child Health and Human Development, criteria includes maternal fever (≥39 °C or ≥38 °C for 30 min) accompanied by one or more of the following [12]:
Fetal tachycardia (>160 bpm for ≥ 10 min)Maternal leukocytosis (WCC >15,000/mm^3^)Purulent fluid from the cervical osBiochemical or microbiologic amniotic fluid results consistent with MIAC
▪The authors noted the current paucity of evidence of current biomarkers and agreed on the critical need for further validation and discovery.


#### 1.1.4. Histological Chorioamnionitis (HCA)

The presence of diffuse leukocyte infiltration into the chorioamniotic membranes. HCA is staged based on the anatomical region of neutrophil infiltration and graded based on the extent of infiltration. The system reported by Redline et al. (2003) [13] is widely used and classifies acute chorioamnionitis into two categories: maternal inflammatory response and fetal inflammatory response (See Table 1).

#### 1.1.5. Fetal Inflammatory Response Syndrome (FIRS)

FIRS, defined by elevation of fetal cord blood plasma interleukin-6 (>11 pg/mg) and histologically by funisitis and chorionic vasculitis is associated with increased risk of multiorgan involvement, neonatal morbidity and mortality [14,15].

### 1.2. Epidemiology/Incidence

The overall worldwide reported incidence of chorioamnionitis varies due to the wide-ranging diagnostic criteria for chorioamnionitis (clinical vs. histological) and generalized criteria for diagnosis. A recent meta-analysis of global incidence of clinical chorioamnionitis reported a pooled incidence of 4.1%; however, rates ranged from 0.6% to 19.7% and there is likely significant under-reporting in developing nations [16]. Within high resource settings such as Singapore, rates are likely on the lower estimate of this range. For example, a recent US based retrospective cohort study of ~9,000,000 term births reported a chorioamnionitis incidence of 1.29% [17]. 

It is well recognized that chorioamnionitis disproportionately affects extremely preterm gestations, with the incidence of chorioamnionitis inversely related to gestational age. For example reported rates of chorioamnionitis at term range from 5 to 12% compared to preterm (22 to 37 weeks gestation) rates of 20–70% [2,18]. Furthermore, Anand et al. reported a 40.7% incidence of chorioamnionitis complicating delivery of very low birth weight babies (<1.5 kg) born at the largest tertiary center in Singapore. Of these deliveries 48% were born prior to 29 completed weeks of gestation [19].

### 1.3. Pathogenesis of the Intra-amniotic Inflammatory Response

Gestational membranes are composed of three layers: the amnion, chorion, and decidua. The decidual layer is the outermost layer and is composed primarily of decidual stromal cells (DSCs) [20]. These cells are believed to respond to infectious agents and so likely play a key role in the pathogenesis of chorioamnionitis [21].

DSCs express pattern recognition receptors (PRRs)—including NOD-like receptors (NLRs) and Toll-like receptors (TLRs). PRR recognize and bind to bacterial pathogen-associated molecular patterns (PAMPs) which results in an inflammatory cascade increasing expression of proinflammatory cytokines such as MCP-1 (monocyte chemo-attractant protein-1), interleukin (IL)-1β, interleukin (IL)-8, interleukin-6 (IL-6) and tumor necrosis factor (TNF)-α. Increased cytokine expression in turn, stimulates the synthesis and release of prostaglandin E2 (PGE2) and matrix metalloproteases (MMPs) [21] (Figure 1 and Table 2). This inflammatory response can cause uterine contractions and structural changes in the cervix, which can lead to cervical ripening and resultant increased risk of preterm birth [22,23]. 

Recent evidence suggests that the inflammatory cascade associated with chorioamnionitis may be activated during periods of cellular stress or injury in the absence of demonstrable infectious triggers and pathogens; this is known as sterile intra-amniotic inflammation [4,24]. For example, Romero et al. demonstrated that, in a population of patients presenting with preterm labor and intact membranes, the incidence of sterile intra-amniotic inflammation (negative AF culture and + AF IL-6 > 2.6 ng/mL) was 26% compared to 11% for microbial associated intra-ammonitic inflammation [4]. 

Many of the pro-inflammatory cytokines outlined in the proposed inflammatory response of chorioamnionitis have become diagnostic targets for detecting and predicting chorioamnionitis and are outlined below. 

Pattern recognition receptors (PRRs) expressed by DSCs include toll-like receptors (TLR1, TLR2, TLR3, TLR4, and TLR6) and nucleotide-binding oligomerization domain receptor 2 (NOD2). These receptors bind to Pathogen-associated molecular pattern molecules (PAMPs) such as peptidoglycan (PGN), lipoteichoic acid (LTA) and lipopolysaccharide (LPS). After binding of PAMPs to PRRs, transcription and translation of proinflammatory cytokines—such as MCP-1 (monocyte chemo-attractant protein-1), interleukin-1β (IL-1β), interleukin-6 (IL-6), interleukin-8 (IL-8), and tumor necrosis factor-α (TNF-α) are increased which in turn results in increased production of prostaglandin E2 (PGE2) and matrix metalloproteinases (MMPs). 

**Table 2 biomedicines-11-02922-t002:** Pattern recognition receptors and the inflammatory response of chorioamnionitis.

Pattern Recognition Receptor	Associated Inflammatory Response
TLR-2	Plays a major role in Gram-positive bacterial recognition. TLR2 hetero-dimerizes with either TLR-1 or TLR-6 and these dimers recognize constituents of Gram-positive bacteria such as peptidoglycan (PGN), and lipoteichoic acid (LTA) [22,25].
TLR-3	Involved in the response to viral infection by recognizing double-stranded RNA [22,25].
TLR-4	Recognizes Gram-negative lipopolysaccharide [22,25].
NOD-2	Recognizes muramyl dipeptide (MDP) which is found in peptidoglycan of both Gram-negative and Gram-positive bacteria [26].
MCP-1	Recruitment of monocytes/macrophages to inflammatory sites [27].
IL-1ß	Increases expression of neutrophil chemotactic and activating chemokines as well enhances MMP expression IL-1β also increases the production of cyclooxygenase (COX)-2 and prostaglandin E2 (PGE2) [28,29].
TNF-α	Produced primarily by macrophages and T cells, a potent proinflammatory cytokine which promotes vasodilation, oedema, leukocyte adhesion and indirectly induces fever [30].
IL-6	Activates T cells and natural killer cells; increases the expression of oxytocin receptors on myometrial cells and oxytocin secretion by myometrial cells [29]
IL-8	Recruits neutrophils; cervical remodeling and rupture of the gestational membrane [31].
MMP	Promotes preterm delivery via decidual, gestational membrane, and cervical extracellular matrix degradation [32].
PGE2	Increases intracellular calcium concentrations in myometrial cells and works synergistically with IL-8 in cervical ripening [32,33].

### 1.4. Infectious Causes of Chorioamnionitis

It is proposed that pathogens can reach the amniotic cavity and cause chorioamnionitis via several different pathways. Most commonly, microbial invasion of the amniotic cavity is believed to occur by retrograde or ascending infection from the lower genital tract [2]. Hematogenous spread and iatrogenic infection complicating invasive procedures such as amniocentesis are less common routes of infection. Retrograde spread from the peritoneum via the fallopian tubes have also been proposed [34]. The common pathogens associated with chorioamnionitis are summarized in Table 3, with polymicrobial infection identified in up to half of cases of clinical chorioamnionitis at term [35]. 

### 1.5. Complications

#### 1.5.1. Maternal Complications

Chorioamnionitis is commonly associated with an increased risk of maternal infectious complications including bacteremia (5% to 10%), endometritis, thromboembolism and pelvic abscess formation (up to threefold increase) [44]. In rare cases chorioamnionitis may result in septic shock and maternal mortality [44]. 

Furthermore, increased risk of abnormal labor, uterine atony (2.5 fold increased risk), post-partum hemorrhage and maternal blood transfusion (2.25 fold increased risk) are associated with chorioamnionitis and are thought to be due to dysfunctional uterine muscle contraction as a result of the maternal inflammatory response [18,44,45]. Interestingly, a study which assessed the association between duration of chorioamnionitis with adverse maternal outcomes found that only uterine atony was related to duration of chorioamnionitis [18].

#### 1.5.2. Fetal/Neonatal Complications

Adverse neonatal outcomes associated with chorioamnionitis include FIRS, early onset neonatal sepsis (EONS), bronchopulmonary dysplasia (BPD), intraventricular hemorrhage (IVH), periventricular leukomalacia (PVL), cerebral palsy, necrotizing enterocolitis (NEC) as well as perinatal death [46,47]. Concerningly, 23–37% of all reported stillbirths are associated with chorioamnionitis [48]. 

#### 1.5.3. Neonatal Sepsis 

Neonatal sepsis, classified as either early-onset (≤72 h after birth) or late-onset (>72 h after birth), is a significant cause of neonatal morbidity and mortality, particularly in preterm infants. A meta-analysis by Beck et al. found that both clinical and histological chorioamnionitis were associated with a three to six-fold increased risk of early onset neonatal sepsis and a 1.35 to 1.75-fold increased odds ratio of late onset neonatal sepsis in preterm infants [49]. 

#### 1.5.4. Bronchopulmonary Dysplasia (BPD)

A meta-analysis of 244,000 infants concluded that both clinical and histological chorioamnionitis was associated with developing bronchopulmonary dysplasia (BPD) [50]. It is postulated that while inflammation of alveoli in utero in the setting of chorioamnionitis causes accelerated lung maturation, this results in changes in the anatomy of the lung such as reduction in the number of alveoli, impaired microvascular development, and thickening of the arteriolar walls, which predisposes the neonate to the development of BPD [50].

It has also been reported that neonates with bronchopulmonary dysplasia had a significantly higher median IL-6 concentration in umbilical cord plasma at birth and elevated umbilical cord plasma IL-6 concentration was a better predictor of the development of bronchopulmonary dysplasia than amniotic fluid IL-6 concentration [51]. This supports the theory that development of fetal inflammatory response (as defined by plasma IL-6 concentration) is associated with inflammation/injury of fetal lungs in utero, which results in bronchopulmonary dysplasia.

#### 1.5.5. Cerebral Palsy and Cystic Periventricular Leukomalacia

The fetal inflammatory response may also result in cerebral white matter injury as inflammatory cytokines released during chorioamnionitis have been suggested to cause cerebral injury via effects on cerebral vasculature that cause cerebral hypoperfusion and a direct toxic effect on the white matter in the fetal brain [52]. This may lead to long term neurological impairment such as cerebral palsy and periventricular leukomalacia [52]. 

This is supported by multiple studies which have linked perinatal brain injury with chorioamnionitis. A meta-analysis of 30 studies reported an association between IAI and development of periventricular leukomalacia (RR 3.0) and cerebral palsy in the preterm (RR 1.9) and term (RR 4.7) neonate [53].

#### 1.5.6. Necrotizing Enterocolitis (NEC)

Garzoni et al. describes a two-hit hypothesis whereby intrauterine infection and intrauterine exposure to inflammatory cytokines prime innate immune cells which results in a subsequent exaggerated inflammatory response associated with NEC when exposed to inflammatory stimuli during labor or postnatally [54]. This association is supported by a meta-analysis (Been et al.) which shows that clinical chorioamnionitis (OR, 1.24; 95% CI, 1.01–1.52) and histological chorioamnionitis with fetal involvement (OR, 3.29; 95% CI, 1.87–5.78) were highly associated with NEC [55]. 

### 1.6. Mode of Delivery 

In the setting of confirmed chorioamnionitis, delivery should be expediated irrespective of gestational age [56]. Evidence suggests that vaginal delivery (including induction of labor) is a safer option for mothers (reduced adverse maternal outcomes) when compared to caesarean section [57]. Furthermore, studies suggest there is no correlation between time from chorioamnionitis diagnosis to delivery and increased risk of poor maternal or neonatal outcomes [56,57]. Therefore, if no obstetric contraindications exist, vaginal delivery should be considered first line management in the setting of confirmed chorioamnionitis. Caesarean section should be reserved for standard obstetric indications. 

## 2. Diagnosis

The accurate and timely diagnosis of chorioamnionitis remains challenging. Given the burden of disease related to chorioamnionitis, the consequences of misdiagnosis, and the need for timely interventions to minimize the risk of preterm labor and fetal injury, the development of rapid and accurate tests for chorioamnionitis is a matter of paramount importance. 

### 2.1. Clinical Signs and Symptoms

The sensitivity and specificity of the clinical chorioamnionitis criteria for detection of MIAC are illustrated in Table 4. 

As represented in Table 4, the diagnostic criteria for clinical chorioamnionitis have variable specificity and sensitivity for MIAC, largely due to the generalized nature of the signs/symptoms and recognized confounding factors present in labor such as fever associated with prostaglandin E2 (PGE2) induction of labor, epidural associated fever [58], fever associated with obstructed labor and use of intrapartum antipyretic medication [11]. Of note, when more than three of the clinical criteria are included, sensitivity and specificity is not improved.

Furthermore, Maki et al. recently reported that the Gibbs clinical criteria for chorioamnionitis had a sensitivity of 15.4%, a specificity of 98.8%, positive likelihood ratio of 13.2 and negative likelihood ratio of 0.86 for detection of IAI. In comparison, the “Suspected Triple I” criteria demonstrated a similar sensitivity of 15.4%, specificity of 100% and negative likelihood ratio of 0.85 [59] for detection of IAI. Given these findings, in patients presenting with suspected chorioamnionitis or Triple I, the clinical criteria should be interpreted cautiously and not solely relied upon to make a diagnosis. 

### 2.2. Maternal Plasma Biomarkers

Despite widespread clinical use, maternal plasma markers such as white blood cell count (WCC), C-reactive protein (CRP) and procalcitonin have poor sensitivity and specificity for chorioamnionitis and there is insufficient evidence to support the use of plasma biomarkers as primary diagnostic tools for chorioamnionitis. 

#### 2.2.1. Full Blood Count

An elevated leukocyte count is commonly associated with bacterial infection and chorioamnionitis, but also many other obstetric conditions such as spontaneous labor and antenatal steroid administration which are commonly prescribed in the setting of preterm prelabor rupture of membranes (PPROM). A recent meta-analysis by Catao Sabogal et al. of four studies (which included two studies complicated by PPROM, one study with preterm labor and one with clinical presentation of chorioamnionitis at term) reported that a leukocyte count of >15,000 leukocytes per mm^3^ had a 53% sensitivity and 73% specificity for detection of HCA. By comparison, a change of >30% in white blood cell count from admission, demonstrated a sensitivity range of 23–63% with a high occurrence of false negatives for the detection of HCA [60]. 

#### 2.2.2. CRP

CRP is an acute phase reactant associated with inflammation related to infection. CRP is secreted by the liver in response to IL6 released by leukocytes and has a half-life of 20 h. Multiple studies and meta-analyses, utilizing a wide range of cut off values (0.3–18.7), have failed to show any accurate diagnostic capabilities of CRP to detect MIAC, HCA or neonatal infection, in the setting of women presenting with suspected chorioamnionitis [61,62,63,64,65] or PPROM [60,66,67]. For example, a meta-analysis of 13 studies (cut ranges varied from >0.3 to 18.7) concluded that CRP had a combined sensitivity of 68.7% and a combined specificity of 77.1% for the detection of HCA, with a positive likelihood ration of 3 and negative likelihood ratio of 0.4 [60]. 

As such, we do not recommend CRP be used as a diagnostic tool, but may be considered as a rule out test to help decide whether patients are high or low risk due to a high negative predictive value, especially in the setting of PPROM. Furthermore, it is recommended that CRP not be repeated earlier than every 48 hrs due to its long half-life. 

#### 2.2.3. Other Cytokine Markers of Inflammation

Studies investigating other maternal plasma biomarkers of inflammation such as IL-1β, IL-2, IL-6, IFN-γ and TNF-α, IL-10, and GCSF have failed to show any clinically diagnostic promise and should be considered experimental at present. For example, Romero et al. recently demonstrated that the cytokine markers commonly associated with inflammation (IL-1β, IL-2, IL-6, IFN-γ and TNF-α) do not accurately distinguish between patients with intrapartum fever without infection, patients in labor and patients with proven microbial associated intra-amniotic infection [68]. Multiple other studies have similarly concluded that maternal inflammatory biomarkers (IL-6 (AUC 0.62)/IL-10 (AUC 0.595)/GCSF (0.602)) lack accuracy in predicting neonatal adverse outcome in the setting of PPROM and therefore should not be used to guide clinical practice [69]. 

#### 2.2.4. Procalcitonin

Procalcitonin is an acute phase reactant and marker of monocyte activity. Multiple studies have demonstrated that procalcitonin in plasma or amniotic fluid is not an accurate predictor of HCA or MIAC and is not suggested for use clinically in the diagnosis of chorioamnionitis [65,70,71]. 

### 2.3. Vaginal Biomarkers

Vaginal biomarkers have been proposed as a diagnostic tool for chorioamnionitis given the minimally invasive nature of collection and multiple studies showing a positive correlation between MIAC and the inflammatory proteins IL-6 and IL-8 in cervicovaginal fluid [72,73]. On the contrary, Cobo et al. demonstrated no significant diagnostic accuracy of cervico-vaginal IL-6 or IL-8 levels [74] for the detection of MIAC, discrepancies between studies could be due to sampling methods of cervico-vaginal fluid and low study population numbers. 

#### 2.3.1. IL-6

Combs et al. analyzed 43 parameters in cervico-vaginal fluid (CVF) of 306 patients with preterm labor but intact membranes, to predict IAI of which IL-6 CVF concentration ≥ 436 pg/mL had the best test performance sensitivity 79%, specificity 78%, PPV 27%, NPV 97%, AUC 0.80 [75]. A study by Jun et al. of women with preterm premature rupture of membranes (PPROM) investigating cervical fluid IL-6 concentration of >350 pg/mL demonstrated a sensitivity of 92% and specificity of 78% to detect MIAC and was associated with higher rates of funisitis, preterm delivery and neonatal morbidity [76]. Whilst another study of women with PPROM by Mikolajcczyk et al. showed vaginal IL-6 concentration of ≥26.8 pg/mL had a high sensitivity (81.08%) and specificity (76.4%) in predicting FIRS in new-borns [77]. A more recent study assessed vaginal IL-6 concentration daily and demonstrated a significant difference in vaginal IL-6 concentration between pregnancies with signs of fetal inflammation and/or early onset sepsis compared to controls [78]. This difference in concentration was significant up to two days before delivery with vaginal IL-6 showing better test performance in predicting inflammation as compared to maternal parameters such as maternal white blood cell count (WBC), serum-concentration of C-reactive protein and IL-6 concentration [78]. 

#### 2.3.2. IL-8

Cervico-vaginal IL-8 levels were also significantly and positively correlated with amniotic fluid IL-8, although was inferior to amniotic fluid IL-8 in predicting risk of MIAC in women with PPROM [79]. 

#### 2.3.3. Other Targets

Protein markers such as neutrophil defensin-1, defensin-2, calgranulin-A, and calgranulin-C have also been associated with inflammation in the amniotic fluid and placenta. Studies have shown correlation between these proteomic biomarkers and stages as well as grades of histological chorioamnionitis [80]. Furthermore, they have been associated with an increased risk of preterm delivery and early onset neonatal sepsis [81]. 

Despite experimental evidence demonstrating the diagnostic potential of vaginal biomarkers, the assays for these inflammatory markers are complex, not readily available, easily contaminated and lack standardization across laboratories. Furthermore, study numbers in the above literature are low and require further investigation. For these reasons, vaginal biomarkers are at present considered experimental only in most countries including Singapore.

### 2.4. Continuous Electronic Fetal Monitoring

One of the recognized indications for continuous electronic fetal monitoring (EFM) is intrapartum maternal pyrexia [56,82]. Loss of or persistently reduced variability, fetal tachycardia, variable decelerations and non-reactive features on EFM, are recognized non-hypoxic related changes associated with chorioamnionitis. However, Ghidini et al. demonstrated that these changes on EFM lack diagnostic accuracy (67%) and specificity (47%) for HCA [83]. Furthermore, Sameshima et al. demonstrated that non-reassuring changes on EFM including late, variable and prolonged decelerations occurred in up to 24% of pregnancies complicated by intrauterine infection and these EFM changes were not associated with an increased incidence of cerebral palsy [84]. To date, no randomized clinical trials have investigated EFM changes in the setting of chorioamnionitis. In summary, when intrauterine infection exists clinicians should exhibit caution interpreting EFM changes with regard to suspected fetal distress, but standard obstetric management should not be precluded. EFM should not be used to help diagnose intrauterine infection due to poor specificity and diagnostic accuracy. 

### 2.5. Amniocentesis

Amniocentesis and amniotic fluid (AF) culture are internationally recognized as the gold standard for antenatal diagnosis of chorioamnionitis. Animal and clinical studies have identified amniotic fluid markers—including cytokines and matrix-degrading enzymes—released during the inflammatory response that might be used for the diagnosis and prognosis of chorioamnionitis [85]. Amongst these, MMP-8 [86,87] and IL-6 [88,89,90] have a relatively high diagnostic rate and can predict preterm delivery and neonatal complications with good sensitivity (Table 5). Amniocentesis-based analyses are however not without their drawbacks, necessitating an invasive sampling procedure that is not amenable to scaling, conveys a small risk of pregnancy loss, and a small risk of iatrogenic infection [91,92]. 

#### 2.5.1. Amniotic Fluid Culture/Gram Stain

AF culture and Gram stain remains the gold standard for antenatal diagnosis of MIAC. A meta-analysis of two studies with 288 patients presenting with preterm labor and intact membranes demonstrated a sensitivity of 65%, specificity of 99%, PPV of 92%, NPV of 95% and AUC of 0.82 (95% CI 0.74 to 0.90) for detection of MIAC [93]. However, Gram stain is limited in its clinical application to patients presenting with subclinical chorioamnionitis and excludes those requiring imminent delivery for any obstetric reason or in preterm labor due to the extended time required for test results. 

#### 2.5.2. IL-6 

A study by Romero et al., of women presenting with threatened preterm labor and intact membranes demonstrated that higher concentrations of amniotic fluid IL-6 were found in MIAC compared to patients with a negative AF culture (IL-6 median 91.2 ng/mL vs. median 0.4 ng/mL, *p* < 0.001) and that an IL-6 concentration > 11.3 ng/mL had a sensitivity of 93.3% and specificity of 91.6% for the detection of MIAC [88]. Amniotic fluid concentrations of IL-6 were also found to be an independent predictor of preterm delivery, amniocentesis-to-delivery interval and neonatal morbidity and mortality [88]. Furthermore a study by Yoon et al. of patients presenting with preterm labor and intact membranes demonstrated that AF IL-6 concentration > 2.6 ng/mL had an AUC of 0.84 for the detection of MIAC and that sterile intra-amniotic inflammation was more common than IAI (21% vs. 10%, *p* < 0.001), indicating the strong prevalence of sterile inflammation in this cohort [3]. It is now commonly accepted that an AF IL-6 of >2.6 ng/mL is diagnostic of intra-amniotic inflammation [3,94].

#### 2.5.3. MMP-8

The positive detection of MMP-8 (>20 ng/mL) in amniotic fluid on a rapid bedside test is associated with a significantly higher rate of intra-amniotic inflammation, proven amniotic fluid infection and histological chorioamnionitis as well as shorter interval to delivery and neonatal morbidity [86,87]. Furthermore, amniotic fluid MMP-8 concentration is correlated with the severity of acute chorioamnionitis [95]. 

However, amniocentesis is not commonly practiced in most institutions worldwide for multiple reasons. Firstly, there is limited clinical utility given a delay of at least 48 h for culture results to be available and as is the case with vaginal biomarkers the assays for these inflammatory biomarkers are not commonly used in most clinical laboratories and remain experimental. Secondly, while there are novel techniques involving the identification of microbial genes [96] for a rapid diagnosis, amniocentesis is highly invasive with procedure-related risks and may not be feasible to perform in patients with severe oligohydramnios following rupture of membranes. Thirdly, patients with intra-amniotic inflammation commonly have negative amniotic cultures but are still at risk of developing adverse outcomes, similar to patients with IAI. 

#### 2.5.4. Glucose Concentration

Meta analysis and individual clinical studies have illustrated that amniotic fluid glucose concentration is an accurate diagnostic tool (AUC 0.86) for the diagnosis of MIAC, as represented in Table 5 [93,97]. When combined with Gram stain the diagnostic accuracy is further increased (AUC 0.92) [93]. Therefore, when an amniocentesis is performed both Gram stain and glucose concentration is recommended.

#### 2.5.5. Leukocyte Esterase

Leukocyte esterase (LE) is an enzyme produced by leukocytes in amniotic fluid which may be rapidly measured with inexpensive reagent strip tests. A positive result indicates the presence of leukocytes associated with bacterial infection [98,99]. When assessed in amniotic fluid LE has a high sensitivity and specificity for MIAC as represented in Table 5. As such, LE should form part of the standard assessment of amniotic fluid for the accurate diagnosis of chorioamnionitis [98].

**Table 5 biomedicines-11-02922-t005:** Sensitivity and specificity of different amniotic fluid markers [2,86,93,98,99,100].

Marker	Reference Range	Sensitivity of MIAC Detection	Specificity of MIAC Detection
Glucose concentration *	<14 mg/dl	85%	87%
Interleukin 6	>7.9 ng/mL	81%	75%
Matrix metalloproteinase	Positive result	90%	80%
White blood cell count	>30/cubic mm	57%	78%
Leukocyte esterase	Positive result	85–91%	95–100%

* Affected by maternal hyperglycemia. It is important to note that the sensitivity and specificity does vary with different reference ranges for glucose.

### 2.6. Investigations in the Setting of Intrauterine Fetal Demise (IUFD)/Stillbirth

When chorioamnionitis results in IUFD or stillbirth, strong consideration should be taken to perform additional investigations including placental and fetal tissue swab/cultures as the causative pathogen may not necessarily be detected by standard investigations. This is particularly evident when IUFD occurs in the second trimester. Previous studies have described the detection of rare causative pathogens such as *Klebsiella pneumoniae* [101] *Citrobacter koseri* [102] and *Kingella kingae* [103] only when these additional investigations have been performed.

### 2.7. Ultrasound

#### 2.7.1. Biophysical Profile (BPP)

A study investigating the diagnostic utility of an abnormal BPP on USS for 166 women with preterm, premature ROM did not accurately predict HCA [83].

#### 2.7.2. Doppler Assessment of the Fetal Umbilical (UA) and Middle Cerebral Arteries (MCA)

At present there is limited evidence to support the use of either UA or MCA doppler assessment in the setting of PPROM for the diagnosis of chorioamnionitis.

A study by Carroll et al. of 69 patients with pregnancies complicated by PPROM demonstrated no association between UA or MCA pulsatility index (PI) and amniotic/fetal blood culture or pH obtained via amniocentesis/cordocentesis [104]. A more recent study of 504 patients with PPROM supported these findings, demonstrating no significant association between amniotic fluid volume, BPP, MCA (PI) or cerebral–placental ratio (CPR) and the development of chorioamnionitis or composite adverse neonatal outcomes (including perinatal death, hypoxic ischemic encephalopathy, sepsis and periventricular leukomalacia) [105].

One small study by Leizer et al. of 50 pregnant patients complicated by PPROM observed an unexpected, significant association between raised MCA PI and histological chorioamnionitis, but not with MCA PSV [106]. This finding should be interpreted with caution given the low study numbers and need for further physiological understanding of how chorioamnionitis may lead to increased resistance (as illustrated by a raised PI) in the fetal MCA.

### 2.8. Future Directions: Non-Invasive Diagnostic Techniques

There is a clear need, but a current lack of non-invasive, highly accurate diagnostic tests of chorioamnionitis. Recently several studies have proposed a number of novel approaches to solve this problem.

A recent Japanese study by Urushiyama et al. of 83 women presenting in preterm labor with intact membranes demonstrated that Next Generation sequencing of vaginal flora from vaginal swabs with machine learning analysis had an area under the curve of 0.85 AUC, sensitivity 71.4% and specificity 82.4% for prediction of HCA [107].

Furthermore, Oh KJ, Lee J, Romero R, et al. demonstrated the potential diagnostic utility of a point of care test utilizing a novel transcervical fluid collector in women with PPROM (between 16–35 weeks). They illustrated that an IL-8 level of >9.5 ng/mL in ammonitic fluid (AF) obtained via this transcervical collector had a 98% sensitivity, 74% specificity, PPV of 80% and NPV of 98% for identification of IAI and was more accurate than a WCC (≥19 cells/mm^3^) in AF obtained via amniocenteses in the same patients [108].

Improvements in imaging quality via USS and MRI have offered up new diagnostic opportunities in obstetrics. It has been recognized that the fetal thymus undergoes involution in response to infection/inflammation. A study by Akasala et al. demonstrated that USS assessment of fetal thymus diameter (<5th centile) was more accurate (sensitivity 91%, specificity 81%, PPV 82%, NPV 91%) than maternal plasma markers (CRP/erythrocyte sedimentation rate) in the detection of HCA in women with PPROM [109]. This finding was supported by a meta-analysis of five studies (including that by Akasala et al.) demonstrating that fetal thymus <5th centile had an OR of 16.02 (95% CI 4.18–61.36) for HCA in the setting of PPROM [110]. However, samples sizes were small, and these findings need further validation.

Imaging via USS or MRI of other fetal organs such as the liver, spleen, adrenals and placenta in relation to diagnosis of intrauterine infection is limited or not validated at present [111]. However, these organs have been demonstrated to be active in the fetal immune response to infection and thus may prove to have some diagnostic utility in the future with advancing imaging techniques and therefore deserve further evaluation.

## 3. Materials and Methods

A literature search was conducted for articles related to chorioamnionitis, intra-uterine infection, intra-uterine inflammation, in the following databases SCOPUS, PUBMED, MEDLINE, EMBASE, Google Scholar. Data from randomized control trials and systematic reviews were prioritized.

## 4. Conclusions

Many of the current diagnostic tests and clinical criteria for IUI outlined in this review rely on detecting a robust and established maternal and fetal inflammatory response. By the time such an inflammatory response is mounted and therefore detected, the fetus and mother are already at significant risk of harm and long-term damage. Therefore, there is a clear and urgent need for further validation of current tests and to develop novel diagnostic tests with high accuracy, that can detect subclinical IUI, and are ideally minimally invasive, that allow clinicians time to enact early management strategies such as antibiotic use or antenatal steroids as well as appropriate delivery planning to minimize the insult/harm to both the mother and vulnerable developing fetus, thereby helping to improve maternal and neonatal outcomes from this devastating condition.

## Figures and Tables

**Figure 1 biomedicines-11-02922-f001:**
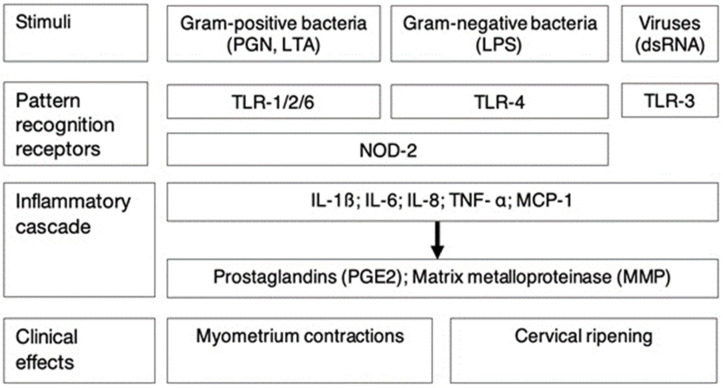
The pathogenesis and associated inflammatory cascade of chorioamnionitis [21,22,23].

**Table 1 biomedicines-11-02922-t001:** Histological classification of acute chorioamnionitis: Staging and Grading [13].

	Maternal Inflammatory Response	Fetal Inflammatory Response
Stage		
Stage 1	Neutrophil infiltration of the subchorionic space (Acute subchorionitis)	Neutrophil infiltration of the chorionic plate vessels and/or umbilical vein (Chorionic vasculitis/umbilical phlebitis)
Stage 2	Neutrophil infiltration of the chorionic plate or chorionic connective tissue and/or amnion (Acute chorioamnionitis)	Neutrophil infiltration in one or both umbilical arteries (Umbilical vasculitis)
Stage 3	Degenerating neutrophils, amnion epithelial necrosis, amnion basement membrane thickening/hypereosinophilia (Necrotizing chorioamnionitis)	Neutrophils and associated debris in a concentric band, ring, or halo around one or more umbilical vessels (Necrotizing funisitis or concentric umbilical perivasculitis)
Grade		
Grade 1 (Mild to moderate)	Individual or small clusters of neutrophils infiltrating the chorion leave, chorionic plate sub chorionic plate, sub chorionic fibrin or amnion	Scattered neutrophilic infiltrate in the subendothelial or intramural portions of any chorionic (or umbilical) vessel
Grade 2 (Severe)	Three or more chorionic micro-abscesses (confluent neutrophils measuring at least 10 × 20 cells in extent) or continuous band of confluent chorionic neutrophils	Near confluent neutrophils in chorionic plate (or umbilical) vessels with attenuation and/or degeneration of vascular smooth muscle cells

**Table 3 biomedicines-11-02922-t003:** Pathogens associated with chorioamnionitis [10,36,37,38,39,40,41,42,43].

Pathogen Origin	Pathogens
Ascending infection	Genital mycoplasma: *Ureaplasma urealyticum, Mycoplasma homini* Anaerobes: *Gardnerella vaginalis, Bacteroides* Aerobes: *Group B Streptococcus* Gram-negative: *Escherichia coli, Syphilis* Fungal organisms: *Candida albicans, Candida tropicalis* and *Candida glabrata ** Viral *HSV*
Hematogenous	Bacterial *Listeria monocytogenes* Parasitic *Toxoplasma Gondii* Viral *Cytomegalovirus*, *adenovirus*, *enterovirus*, *respiratory syncytial virus*, *Epstein–Barr virus*, *Rubella*, *COVID-19, Zika virus*

* These infections have been reported in women with in vitro fertilization pregnancies, in those with retained intrauterine contraceptive devices, following amniocentesis, and in those with prolonged rupture of membranes [37].

**Table 4 biomedicines-11-02922-t004:** Sensitivity and specificity of the clinical chorioamnionitis criteria for detection of MIAC [7].

Clinical Sign/Symptom	Sensitivity MIAC Detection	Specificity MIAC Detection
Maternal fever (temperature > 37.8 °C)	42%	86.5%
Maternal tachycardia (heart rate > 100 beats/min)	88%	5%
Uterine tenderness	12%,	95%
Foul-smelling amniotic fluid/vaginal discharge	8%	95%
Fetal tachycardia (heart rate > 160 beats/min);	80%	30%
Maternal leukocytosis (leukocyte count > 15,000 cells/mm^3^)	76%	30%
Any three or more of above criteria	56%	55%

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
