# Peer review of "Chorioamnionitis: An Update on Diagnostic Evaluation"

_biomedicines, 2023, doi:10.3390/biomedicines11112922_

Round 1

Reviewer 1 Report

Comments and Suggestions for Authors Dear authors, I read with great interest the manuscript, which falls within the aim of this Journal. In my honest opinion, the topic is interesting enough to attract the readers’ attention. Nevertheless, authors should clarify some points and improve the discussion, as suggested below. Authors should consider the following recommendations: I suggest to refer to the updated literature on eventually correlation between chorioamnionitis and other infectios as COVID or ZIKA VIRUS  and to the really needed of NIPT TEST .   I SUGGEST YOU TO READ AND CITE THESE ARTICLES: Symptomatic COVID-19 in Pregnancy: Hospital Cohort Data between May 2020 and April 2021, Risk Factors and Medicolegal ImplicationsCongenital   Zika Syndrome: Genetic Avenues for Diagnosis and Therapy, Possible Management and Long-Term Outcomes.     Contrast Agents during Pregnancy: Pros and Cons When Really Needed   Chorioamnionitis or intrauterine inflammation is a frequent cause of preterm birth. Chorioamnionitis can affect almost every organ of the developing fetus. Multiple microbes have been implicated to cause chorioamnionitis, but "sterile" inflammation appears to be more common.

Comments on the Quality of English Language

Minor editing of English language required

Author Response

20/10/2023

Biomedicines Manuscript 2633911 - Chorioamnionitis: An Update on Diagnostic Evaluation

Response to Reviewers

Dear Biomedicines,

Thank you for the opportunity to submit a revised draft of the manuscript “Chorioamnionitis: An Update on Diagnostic Evaluation” for publication in the Journal Biomedicines.  We appreciate the time and effort that you and the reviewers dedicated to providing valuable feedback on our manuscript and are grateful for the insightful comments on and improvements to our paper. We have incorporated most of the suggestions made by the reviewers. Those changes are highlighted within the manuscript. Please see below, in blue, for point – by – point response to the reviewers’ comments and concerns.   All page numbers refer to the revised manuscript file.  

Reviewer 1

  1. Comment from reviewer: “I suggest to refer to the updated literature on eventually correlation between chorioamnionitis and other infections as COVID or ZIKA VIRUS ... I SUGGEST YOU TO READ AND CITE THESE ARTICLES”

Thankyou for this suggestion, we have incorporated both COVID and Zika viruses into the manuscript highlighted in Table 3 with references included as suggested.

  1. Comment from reviewer to incorporate the article “Contrast Agents during Pregnancy: Pros and Cons When Really Needed”

Thank you for the suggestion to include this article as part of the manuscript, we have however chosen not to included this reference as there is no discussion or reference to chorioamnionitis within this paper and therefore do not think it is appropriate to include in “Chorioamnionitis: An Update on Diagnostic Evaluation”.  

  1. Comment from reviewer “and to the really needed of NIPT TEST”

Thank you for this suggestions, however we believe that we do address the need for non invasive testing in  2.8 Future directions: Non-invasive diagnostic techniques line 562:

“There is a clear need but current lack of non-invasive, highly accurate diagnostic tests of chorioamnionitis. Recently several studies have proposed a number of novel approaches to solve this problem”

Reviewer 2 Report

Comments and Suggestions for Authors

The paper "Chorioamnionitis: An Update on Diagnostic Evaluation" is overall well written and the the related difficulties nicely discussed. 

The Authors should mention the fatal outcome of chorioamnionitis and funisitis, even in unusual infections, that can occur, especially in the second trimester. 

The following papers are an example: 

Bonasoni MP, Palicelli A, Dalla Dea G, Comitini G, Nardini P, Vizzini L, Russello G, Bardaro M, Carretto E. Klebsiella pneumoniaeChorioamnionitis: An Underrecognized Cause of Preterm Premature Rupture of Membranes in the Second Trimester. Microorganisms. 2021 Jan 3;9(1):96. doi: 10.3390/microorganisms9010096. PMID: 33401648; PMCID: PMC7824054.

Bonasoni MP, Comitini G, Pati M, Russello G, Vizzini L, Bardaro M, Pini P, Marrollo R, Palicelli A, Dalla Dea G, Carretto E. Second Trimester Fetal Loss Due to Citrobacter koseri Infection: A Rare Cause of Preterm Premature Rupture of Membranes (PPROM). Diagnostics (Basel). 2022 Jan 10;12(1):159. doi: 10.3390/diagnostics12010159. PMID: 35054326; PMCID: PMC8774530.

Bonasoni MP, Palicelli A, Dalla Dea G, Comitini G, Pazzola G, Russello G, Bertoldi G, Bardaro M, Zuelli C, Carretto E. Kingella kingae Intrauterine Infection: An Unusual Cause of Chorioamnionitis and Miscarriage in a Patient with Undifferentiated Connective Tissue Disease. Diagnostics (Basel). 2021 Feb 4;11(2):243. doi: 10.3390/diagnostics11020243. PMID: 33557386; PMCID: PMC7914692.

Moreover, in these papers, the infective diagnosis was made on fetal tissues and placenta (sub amniotic swab and tissue). 

The Authors should mention the possibility of these ancillary tools in case of fatal outcome in order to identify the etiologic agent. 

Comments on the Quality of English Language

The English is fine. 

Author Response

20/10/2023

Biomedicines Manuscript 2633911 - Chorioamnionitis: An Update on Diagnostic Evaluation

Response to Reviewers

Dear Biomedicines,

Thank you for the opportunity to submit a revised draft of the manuscript “Chorioamnionitis: An Update on Diagnostic Evaluation” for publication in the Journal Biomedicines.  We appreciate the time and effort that you and the reviewers dedicated to providing valuable feedback on our manuscript and are grateful for the insightful comments on and improvements to our paper. We have incorporated most of the suggestions made by the reviewers. Those changes are highlighted within the manuscript. Please see below, in blue, for point – by – point response to the reviewers’ comments and concerns.   All page numbers refer to the revised manuscript file.  

Kind Regards,

Reviewer 2:

  1. Comment from reviewer “The Authors should mention the fatal outcome of chorioamnionitis and funisitis, even in unusual infections, that can occur, especially in the second trimester. The following papers are an example: “

Bonasoni MP, Palicelli A, Dalla Dea G, Comitini G, Nardini P, Vizzini L, Russello G, Bardaro M, Carretto E. Klebsiella pneumoniaeChorioamnionitis: An Underrecognized Cause of Preterm Premature Rupture of Membranes in the Second Trimester. Microorganisms. 2021 Jan 3;9(1):96. doi: 10.3390/microorganisms9010096. PMID: 33401648; PMCID: PMC7824054.  

Bonasoni MP, Comitini G, Pati M, Russello G, Vizzini L, Bardaro M, Pini P, Marrollo R, Palicelli A, Dalla Dea G, Carretto E. Second Trimester Fetal Loss Due to Citrobacter koseri Infection: A Rare Cause of Preterm Premature Rupture of Membranes (PPROM). Diagnostics (Basel). 2022 Jan 10;12(1):159. doi: 10.3390/diagnostics12010159. PMID: 35054326; PMCID: PMC8774530.

Bonasoni MP, Palicelli A, Dalla Dea G, Comitini G, Pazzola G, Russello G, Bertoldi G, Bardaro M, Zuelli C, Carretto E. Kingella kingae Intrauterine Infection: An Unusual Cause of Chorioamnionitis and Miscarriage in a Patient with Undifferentiated Connective Tissue Disease. Diagnostics (Basel). 2021 Feb 4;11(2):243. doi: 10.3390/diagnostics11020243. PMID: 33557386; PMCID: PMC7914692.

Moreover, in these papers, the infective diagnosis was made on fetal tissues and placenta (sub amniotic swab and tissue). 

The Authors should mention the possibility of these ancillary tools in case of fatal outcome in order to identify the etiologic agent. 

Thank you for these important suggestions. We have included the following updates to the manuscript (highlighted in text)

“Concerningly, 23 - 37% of all reported stillbirths are associated with chorioamnionitis[49]” highlighted at line 240. 

We have also included the paragraph below referencing to the above recommended papers and ancillary tools in the case of fatal outcome (highlighted at line 524):

“2.6 Investigations in the setting of intrauterine fetal demise (IUFD) / stillbirth

When chorioamnionitis results in IUFD or stillbirth, strong consideration should be taken to perform additional investigations including placental and fetal tissue swab/cultures as the causative pathogen may not necessarily be detected by standard investigations. This is particularly evident when IUFD occurs in the second trimester.  Previous studies have described the detection of pathogens such as Klebsiella pneumoniae [98] Citrobacter koseri [99] and Kingella kingae [100] only when these additional investigations have been performed. “

Reviewer 3 Report

Comments and Suggestions for Authors

I think this review is concise and meaningful. So, I have only one request. I think the possibility of selecting cesarean delivery will be increased. I want you to provide the comments for this situation.

Author Response

20/10/2023

Biomedicines Manuscript 2633911 - Chorioamnionitis: An Update on Diagnostic Evaluation

Response to Reviewers

Dear Biomedicines,

Thank you for the opportunity to submit a revised draft of the manuscript “Chorioamnionitis: An Update on Diagnostic Evaluation” for publication in the Journal Biomedicines.  We appreciate the time and effort that you and the reviewers dedicated to providing valuable feedback on our manuscript and are grateful for the insightful comments on and improvements to our paper. We have incorporated most of the suggestions made by the reviewers. Those changes are highlighted within the manuscript. Please see below, in blue, for point – by – point response to the reviewers’ comments and concerns.   All page numbers refer to the revised manuscript file.  

Kind Regards,

Reviewer 3:

  1. Reviewer comment: “I think this review is concise and meaningful. So, I have only one request. I think the possibility of selecting cesarean delivery will be increased. I want you to provide the comments for this situation.”

Thank you for the suggestion of including commentary on mode of delivery in the setting of chorioamnionitis. We have included this suggestion in the manuscript at line 288 (highlighted) as below:

“1.7 Mode of delivery

In the setting of confirmed chorioamnionitis delivery should be expediated irrespective of gestational age[57].  Evidence suggests that vaginal delivery (including induction of labour) is a safer option for mothers (reduced adverse maternal outcomes) when compared to caesarean section[58]. Furthermore, studies suggest there is no correlation between time from chorioamnionitis diagnosis to delivery and increased risk of poor maternal or neonatal outcomes [57, 58]. Therefore, if no obstetric contraindications exist, vaginal delivery should be considered first line management in the setting of confirmed chorioamnionitis. Caesarean section should be reserved for standard obstetric indications. “

Reviewer 4 Report

Comments and Suggestions for Authors

The article concerns one of the most important causes of premature birth, which is chorioamnionitis. The Authors focused on its diagnosis. They undertook the difficult task of presenting currently available methods for diagnosing this pathology of pregnancy, which has been a difficult challenge for obstetricians and neonatologists for many years.

The article is a good compendium of knowledge for specialists.

In the introduction, please include the concept of fetal inflammatory response syndrome (FIRS).

It seems advisable to organize diagnostic methods with a clear division into:

1. laboratory methods – newer methods should also be mentioned in this part.

Please briefly describe in the text (and not only in the table):

- Glucose concentration

- Leukocyte esterase activity

2. Non-laboratory methods:

- Cardiotocography - please consider parameters other than those mentioned, such as minimal or moderate baseline variability and late decelerations.

- Ultrasound - describing not only the biophysical profile and the fetal thymus, but also the fetal liver, adrenal glands and spleen, as well as the placenta. Please mention the Doppler ultrasound examination. In the article, the Authors mention PPROM - please refer to this pathology in relation to the ultrasound examination.

- Magnetic resonance imaging.

Please expand all abbreviations the first time you use them (for example PPROM).

Comments on the Quality of English Language

Minor errors, e.g. a/the in line 34.

Author Response

20/10/2023

Biomedicines Manuscript 2633911 - Chorioamnionitis: An Update on Diagnostic Evaluation

Response to Reviewers

Dear Biomedicines,

Thank you for the opportunity to submit a revised draft of the manuscript “Chorioamnionitis: An Update on Diagnostic Evaluation” for publication in the Journal Biomedicines.  We appreciate the time and effort that you and the reviewers dedicated to providing valuable feedback on our manuscript and are grateful for the insightful comments on and improvements to our paper. We have incorporated most of the suggestions made by the reviewers. Those changes are highlighted within the manuscript. Please see below, in blue, for point – by – point response to the reviewers’ comments and concerns.   All page numbers refer to the revised manuscript file.  

Kind Regards,

Reviewer 4:

  1. Reviewer comment: “In the introduction, please include the concept of fetal inflammatory response syndrome (FIRS).”

Thank you for this suggestion, we have moved the definition of FIRS up to the introduction as requested at line 101 (highlighted in text)

“1.2.5 Fetal Inflammatory Response Syndrome (FIRS)

  • FIRS, defined by elevation of fetal cord blood plasma interlekin-6 (>11pg/mg) and histologically by funisitis and chorionic vasculitis is associated with increased risk of multiorgan involvement, neonatal morbidity and mortality[14, 15]. “

  1. Reviewer comment 2: “It seems advisable to organize diagnostic methods with a clear division into:
    1. laboratory methods – newer methods should also be mentioned in this part.
    2. Non-laboratory methods:

Thank you for the suggestion for how to organise and structure this review paper. However, we have chosen to keep the current structure as we believe organising the paper into laboratory methods and non-laboratory methods would lead to too many subcategories under each of these headings which makes reading the review difficult. 

  1. Please briefly describe in the text (and not only in the table):

- Glucose concentration

- Leukocyte esterase activity

Thankyou for this suggestions, we believe this is an important addition to the manuscript, please see below additional text. 

Line 501 highlighted in text:

2.5.4 Glucose concentration

Meta analysis and individual clinical studies have illustrated that amniotic fluid glucose concentration is an accurate diagnostic tool (AUC 0.86) for the diagnosis of MIAC, as represented in Table 5 [94, 98].  When combined with gram stain the diagnostic accuracy is further increased (AUC 0.92)[94].  Therefore, when an amniocentesis is performed both gram stain and glucose concentration is recommended. 

Line 508 highlighted in text:

2.5.5 Leukocyte esterase

Leukocyte esterase (LE) is an enzyme produced by leukocytes in amniotic fluid which may be rapidly measured on inexpensive reagent strip tests.  A positive result indicates the presence of leukocytes associated with bacterial infection [99, 100].  When assessed in amniotic fluid LE has a high sensitivity and specificity for MIAC as represented in Table 5. As such LE should form part of the standard assessment of amniotic fluid for the accurate diagnosis of chorioamnionitis [99]. 

  1. Reviewer comment : “Cardiotocography - please consider parameters other than those mentioned, such as minimal or moderate baseline variability and late decelerations.”

Thank you for this excellent suggestion. We have included further discussion around the use of cardiotocography under the section 2.4 Continuous Electronic Fetal Monitoring at line 428 (changes highlighted in text)

“One of the recognised indications for continuous electronic fetal monitoring (EFM) is intrapartum maternal pyrexia [57, 83].  Loss of or persistently reduced variability, fetal tachycardia, variable decelerations and non-reactive features on EFM, are recognised non-hypoxic related changes associated with chorioamnionitis. However, Ghidini et al. demonstrated that these changes on EFM lack diagnostic accuracy (67%) and specificity (47%) for HCA [84].  Furthermore, Sameshima et al. demonstrated that non-reassuring changes on EFM including late, variable and prolonged decelerations occurred in up to 24% of pregnancies complicated by intrauterine infection and these EFM changes were not associated with an increased incidence of cerebral palsy [85]. To date, no randomised clinical trials have investigated EFM changes in the setting of chorioamnionitis.  In summary, when intrauterine infection exists clinicians should exhibit caution interpreting EFM changes with regards to suspected fetal distress, but standard obstetric management should not be precluded. EFM should not be used to help diagnose intrauterine infection due to poor specificity and diagnostic accuracy.”    

  1. Reviewer comment : “Ultrasound - describing not only the biophysical profile and the fetal thymus, but also the fetal liver, adrenal glands and spleen, as well as the placenta.” And “Magnetic resonance imaging.”

Thank you for this suggestion we have included a discussion on these point in the section under Future directions: Non-invasive diagnostic techniques at line 588 (highlighted) and copied below:

“Imaging via USS or MRI of other fetal organs such as the liver, spleen, adrenals and placenta in relation to diagnosis of intrauterine infection is limited or not validated at present[107].  However, these organs have been demonstrated to be active in the fetal immune response to infection and thus may prove to have some diagnostic utility in the future with advancing imaging techniques and deserve further evaluation.”

  1. Reviewer comment: Please mention the Doppler ultrasound examination. In the article, the Authors mention PPROM - please refer to this pathology in relation to the ultrasound examination.

Thank you for this suggestion we have included a discussion on Doppler USS assessment and PPROM in relation to ultrasound in the added section “2.7.2 Doppler assessment of the fetal umbilical (UA) and middle cerebral arteries (MCA)” at line 539 (highlighted in text) and included below which addresses these issues:

“2.7.2 Doppler assessment of the fetal umbilical (UA) and middle cerebral arteries (MCA)

At present there is limited evidence to support the use of either UA or MCA doppler assessment in the setting of PPROM for the diagnosis of chorioamnionitis.

A study by Carroll et al. of 69 patients with pregnancies complicated by PPROM demonstrated no association between UA or MCA pulsatility index (PI) and amniotic/fetal blood culture or pH obtained via amniocentesis/cordocentesis [105].  A more recent study of 504 patients with PPROM supported these findings, demonstrating no significant association between amniotic fluid volume, BPP, MCA (PI) or Cerebral-Placental Ratio (CPR) and the development of chorioamnionitis or composite adverse neonatal outcome (including perinatal death, hypoxic ischemic encephalopathy, sepsis and periventricular leukomalacia) [106].

One small study by Leizer et al. of 50 pregnant patients complicated by PPROM observed an unexpected, significant association between raised MCA PI and histological chorioamnionitis but not with MCA PSV [107]. This finding should be interpreted with caution given the low study numbers and need for further physiological understanding of how chorioamnionitis may lead to increased resistance (as illustrated by a raised PI) in the fetal MCA.”

  1. Reviewer comment: Please expand all abbreviations the first time you use them (for example PPROM).

Thankyou for this suggestion – we have checked and the first use of PRROM is at line 346 which is accompanied by the fully expanded definition.  We have also ensured that all acronyms elsewhere in the text are fully expanded. 

Grammar suggestions:

Thank you for the grammar suggestion we have removed “the” at  Line 34

Round 2

Reviewer 1 Report

Comments and Suggestions for Authors

now the paper in my opinion is suitable for pubblication